# Zero-Shot Self-Supervised Learning for MRI Reconstruction

**Burhaneddin Yaman**[†‡]**, Seyed Amir Hossein Hosseini**[†‡]**, Mehmet Akçakaya**[†‡]
† Department of Electrical&Computer Engineering, University of Minnesota
‡ Center for Magnetic Resonance Research, University of Minnesota
{yaman013, hosse049, akcakaya}@umn.edu

## Abstract

Deep learning (DL) has emerged as a powerful tool for accelerated MRI reconstruction, but often necessitates a database of fully-sampled measurements for training. Recent self-supervised and unsupervised learning approaches enable training without fully-sampled data. However, a database of undersampled measurements may not be available in many scenarios, especially for scans involving contrast or translational acquisitions in development. Moreover, recent studies show that database-trained models may not generalize well when the unseen measurements differ in terms of sampling pattern, acceleration rate, SNR, image contrast, and anatomy. Such challenges necessitate a new methodology to enable subject-specific DL MRI reconstruction without external training datasets, since it is clinically imperative to provide high-quality reconstructions that can be used to identify lesions/disease for *every individual*. In this work, we propose a zero-shot self-supervised learning approach to perform subject-specific accelerated DL MRI reconstruction to tackle these issues. The proposed approach partitions the available measurements from a single scan into three disjoint sets. Two of these sets are used to enforce data consistency and define loss during training for self-supervision, while the last set serves to self-validate, establishing an early stopping criterion. In the presence of models pre-trained on a database with different image characteristics, we show that the proposed approach can be combined with transfer learning for faster convergence time and reduced computational complexity.

## 1 Introduction

Magnetic resonance imaging (MRI) is a non-invasive, radiation-free medical imaging modality that provides excellent soft tissue contrast for diagnostic purposes. However, lengthy acquisition times in MRI remain a limitation. Accelerated MRI techniques acquire fewer measurements at a sub-Nyquist rate, and use redundancies in the acquisition system or the images to remove the resulting aliasing artifacts during reconstruction. In clinical MRI systems, multi-coil receivers are used during data acquisition. Parallel imaging (PI) is the most clinically used method for accelerated MRI, and exploits the redundancies between these coils for reconstruction (Pruessmann et al., 1999; Griswold et al., 2002). Compressed sensing (CS) is another conventional accelerated MRI technique that exploits the compressibility of images in sparsifying transform domains (Lustig et al., 2007), and is commonly used in combination with PI. However, PI and CS may suffer from noise and residual artifacts at high acceleration rates (Robson et al., 2008; Sandino et al., 2020).

Recently, deep learning (DL) methods have emerged as an alternative for accelerated MRI due to their improved reconstruction quality compared to conventional approaches (Hammernik et al., 2018; Knoll et al., 2020b; Akçakaya et al., 2022). Particularly, physics-guided deep learning reconstruction (PG-DLR) approaches have gained interest due to their robustness and improved performance (Hammernik et al., 2018; Hosseini et al., 2020b). PG-DLR explicitly incorporates the physics of the data acquisition system into the neural network via a procedure known as algorithm unrolling (Monga et al., 2021). This is done by unrolling iterative optimization algorithms that alternate between data consistency (DC) and regularization steps for a fixed number of iterations. Subsequently, PG-DLR methods are trained in a supervised manner using large databases of fully-sampled measurements (Hammernik et al., 2018; Aggarwal et al., 2019). More recently, self-supervised learning

has shown that reconstruction quality similar to supervised PG-DLR can be achieved while training on a database of only undersampled measurements (Yaman et al., 2020).

While such database learning strategies offer improved reconstruction quality, acquisition of large datasets may often be infeasible. In some MRI applications involving time-varying physiological processes, dynamic information such as time courses of signal changes, contrast uptake or breathing patterns may differ substantially between subjects, making it difficult to generate high-quality databases of sufficient size for the aforementioned strategies. Furthermore, database training, in general, brings along concerns about robustness and generalization (Eldar et al., 2017; Knoll et al., 2020c). In MRI reconstruction, this may exhibit itself when there are mismatches between training and test datasets in terms of image contrast, sampling pattern, SNR, vendor, and anatomy. While it is imperative to have high-quality reconstructions that can be used to correctly identify lesions/disease for *every individual*, the fastMRI transfer track challenge shows that pretrained models fail to generalize when applied to patients/scans with different distribution or acquisition parameters, with potential for misdiagnosis (Muckley et al., 2021). Finally, training datasets may lack examples of rare and/or subtle pathologies, increasing the risk of generalization failure (Knoll et al., 2019; 2020c).

In this work, we tackle these challenges associated with database training, and propose a zero-shot self-supervised learning (ZS-SSL) approach, which performs subject-specific training of PG-DLR without any external training database. Succinctly, ZS-SSL partitions the acquired measurements into three types of disjoint sets, which are respectively used only in the PG-DLR neural network, in defining the training loss, and in establishing a stopping strategy to avoid overfitting. Thus, our training is both self-supervised and self-validated. In cases where a database-pretrained network is available, ZS-SSL leverages transfer learning (TL) for improved reconstruction quality and reduced computational complexity.

Our contributions can be summarized as follows:

- We propose a zero-shot self-supervised method for learning subject-specific DL MRI reconstruction from a single undersampled dataset without any external training database.
- We provide a well-defined methodology for determining stopping criterion to avoid overfitting in contrast to other single-image training approaches (Ulyanov et al., 2018).
- We apply the proposed zero-shot learning approach to knee and brain MRI datasets, and show its efficacy in removing residual aliasing and banding artifacts compared to supervised database learning.
- We show our ZS-SSL can be combined with with TL in cases when a database-pretrained model is available to reduce computational costs.
- We show that our zero-shot learning strategies address robustness and generalizability issues of trained supervised models in terms of changes in sampling pattern, acceleration rate, contrast, SNR, and anatomy at inference time.

## 2 BACKGROUND AND RELATED WORK

### 2.1 ACCELERATED MRI ACQUISITION MODEL

In MRI, raw measurement data is acquired in the frequency domain, also known as k-space. In current clinical MRI systems, multiple receiver coils are used, where each is sensitive to different parts of the volume. In practice, MRI is accelerated by taking fewer measurements, which are characterized by an undersampling mask that specifies the acquired locations in k-space. For a multi-coil MRI acquisition, the forward model is given as

$$\mathbf{y}_i = \mathbf{P}_\Omega \mathcal{F} \mathbf{C}_i \mathbf{x} + \mathbf{n}_i, \ \ i \in \{1, \dots, n_c\}, \tag{1}$$

where $\mathbf{x}$ is the underlying image, $\mathbf{y}_i$ is the acquired data for the $i^{\text{th}}$ coil, $\mathbf{P}_\Omega$ is the masking operator for undersampling pattern $\Omega$, $\mathcal{F}$ is the Fourier transform, $\mathbf{C}_i$ is a diagonal matrix characterizing the $i^{\text{th}}$ coil sensitivity, $\mathbf{n}_i$ is measurement noise for $i^{\text{th}}$ coil, and $n_c$ is the number of coils (Pruessmann et al., 1999). This system can be concatenated across the coil dimension for a compact representation

$$\mathbf{y}_\Omega = \mathbf{E}_\Omega \mathbf{x} + \mathbf{n}, \tag{2}$$

where $\mathbf{y}_\Omega$ is the acquired undersampled measurements across all coils, $\mathbf{E}_\Omega$ is the forward encoding operator that concatenates $\mathbf{P}_\Omega \mathcal{F} \mathbf{C}_i$ across $i \in \{1, \dots, n_c\}$. The general inverse problem for

accelerated MRI is given as

$$\arg\min_{\mathbf{x}} \|\mathbf{y}_\Omega - \mathbf{E}_\Omega \mathbf{x}\|_2^2 + \mathcal{R}(\mathbf{x}), \tag{3}$$

where the $\|\mathbf{y}_\Omega - \mathbf{E}_\Omega \mathbf{x}\|_2^2$ term enforces consistency with acquired data (DC) and $\mathcal{R}(\cdot)$ is a regularizer.

## 2.2 PG-DLR WITH ALGORITHM UNROLLING

Several optimization methods are available for solving the inverse problem in (3) (Fessler, 2020). Variable-splitting via quadratic penalty is one of the approaches that can be employed to cast Eq. (3) into two sub-problems as

$$\mathbf{z}^{(i)} = \arg\min_{\mathbf{z}} \mu \|\mathbf{x}^{(i-1)} - \mathbf{z}\|_2^2 + \mathcal{R}(\mathbf{z}), \tag{4a}$$

$$\mathbf{x}^{(i)} = \arg\min_{\mathbf{x}} \|\mathbf{y}_\Omega - \mathbf{E}_\Omega \mathbf{x}\|_2^2 + \mu \|\mathbf{x} - \mathbf{z}^{(i)}\|_2^2, \tag{4b}$$

where $\mu$ is the penalty parameter, $\mathbf{z}^{(i)}$ is an intermediate variable and $\mathbf{x}^{(i)}$ is the desired image at iteration $i$. In PG-DLR, an iterative algorithm, as in (4a) and (4b) is unrolled for a fixed number of iterations (Liang et al., 2020). The regularizer sub-problem in Eq. (4a) is implicitly solved with neural networks and the DC sub-problem in Eq. (4b) is solved via linear methods such as gradient descent (Hammernik et al., 2018) or conjugate gradient (CG) (Aggarwal et al., 2019).

There have been numerous works on PG-DLR for accelerated MRI (Schlemper et al., 2018; Hammernik et al., 2018; Aggarwal et al., 2019; Liang et al., 2020; Yaman et al., 2020). Most of these works vary from each other on the algorithms used for DC and neural networks employed in the regularizer units. However, all these works require a large database of training samples.

## 2.3 SUPERVISED LEARNING FOR PG-DLR

In supervised PG-DLR, training is performed using a database of fully-sampled reference data. Let $\mathbf{y}_{\text{ref}}^n$ be the fully-sampled k-space for subject $n$ and $f(\mathbf{y}_\Omega^n, \mathbf{E}_\Omega^n; \boldsymbol{\theta})$ be the output of the unrolled network for under-sampled k-space $\mathbf{y}_\Omega^n$, where the network is parameterized by $\boldsymbol{\theta}$. End-to-end training minimizes (Knoll et al., 2020b; Yaman et al., 2020)

$$\min_{\boldsymbol{\theta}} \frac{1}{N} \sum_{n=1}^{N} \mathcal{L}(\mathbf{y}_{\text{ref}}^n, \ \mathbf{E}_{\text{full}}^n f(\mathbf{y}_\Omega^n, \mathbf{E}_\Omega^n; \boldsymbol{\theta})), \tag{5}$$

where $N$ is the number of samples in the training database, $\mathbf{E}_{\text{full}}^n$ is the fully-sampled encoding operator that transform network output to k-space and $\mathcal{L}(\cdot, \cdot)$ is a loss function.

## 2.4 SELF-SUPERVISED LEARNING FOR PG-DLR

Unlike supervised learning, self-supervised learning enables training without fully-sampled data by only utilizing acquired undersampled measurements (Yaman et al., 2020). A masking approach is used for self-supervision in this setting, where a subset $\Lambda \subset \Omega$ is set aside for checking prediction performance/loss calculation, while the remainder of points $\Theta = \Omega \backslash \Lambda$ are used in the DC units of the PG-DLR network. End-to-end training is performed using the loss function

$$\min_{\boldsymbol{\theta}} \frac{1}{N} \sum_{n=1}^{N} \mathcal{L}\Big(\mathbf{y}_\Lambda^n, \ \mathbf{E}_\Lambda^n \big(f(\mathbf{y}_\Theta^n, \mathbf{E}_\Theta^n; \boldsymbol{\theta})\big)\Big). \tag{6}$$

# 3 ZERO-SHOT SELF-SUPERVISED LEARNING FOR PG-DLR

As discussed in Section 1, lack of large datasets in numerous MRI applications, as well as robustness and generalizability issues of pretrained models pose a challenge for the clinical translation of DL reconstruction methods. Hence, subject-specific reconstruction is desirable in clinical practice, since it is critical to achieve a reconstruction quality that can be used for correctly diagnosing every patient. While the conventional self-supervised masking strategy, as in (Yaman et al., 2020) can be applied for subject-specific learning, it leads to overfitting unless the training is stopped early (Hosseini et al., 2020a). This is similar to other single-image learning strategies, such as the deep image prior (DIP)

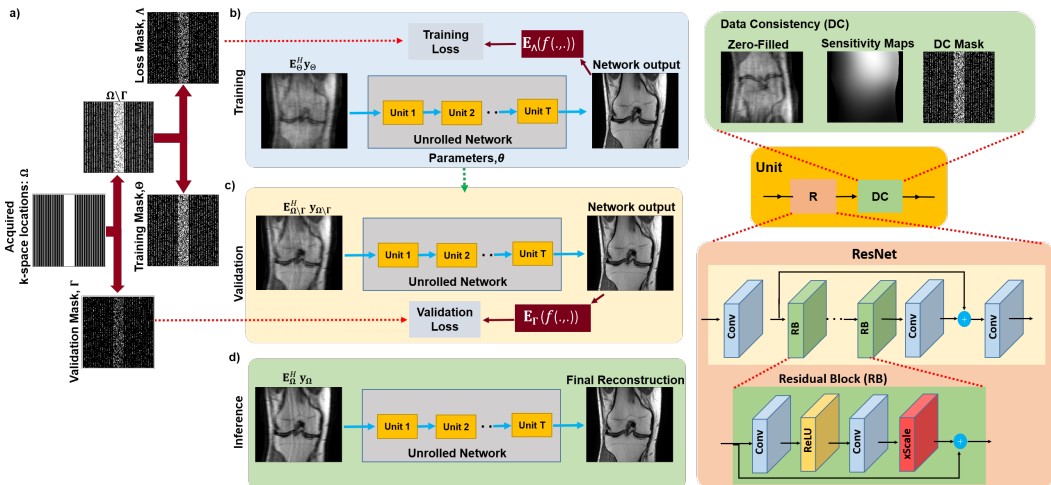

Figure 1: An overview of the proposed zero-shot self-supervised learning approach. a) Acquired measurements for the single scan are partitioned into three sets: a training ($\Theta$) and loss mask ($\Lambda$) for self-supervision, and a self-validation mask for automated early stopping ($\Gamma$). b) The parameters, $\boldsymbol{\theta}$, of the unrolled MRI reconstruction network are updated using $\Theta$ and $\Lambda$ in the data consistency (DC) units of the unrolled network and for defining loss, respectively. c) Concurrently, a k-space validation procedure is used to establish the stopping criterion by using $\Omega\backslash\Gamma$ in the DC units and $\Gamma$ to measure a validation loss. d) Once the network training has been stopped due to an increasing trend in the k-space validation loss, the final reconstruction is performed using the relevant learned network parameters and all the acquired measurements in the DC unit.

or zero-shot super-resolution (Ulyanov et al., 2018; Shocher et al., 2018). DIP-type approaches shows that an untrained neural network can successfully perform instance-specific image restoration tasks such as denoising, super-resolution, inpainting without any training data. However, such DIP-type techniques requires an early stopping for avoiding over-fitting, which is typically done with a manual heuristic selection (Ulyanov et al., 2018; Hosseini et al., 2020a; Darestani et al., 2021). While this may work in a research setting, having a well-defined automated early stopping criterion is critical to fully harness the potential of subject-specific DL MRI reconstruction in practice.

Early stopping regularization in database-trained setting is conventionally motivated through the bias-variance trade-off, in which a validation set is used as a proxy for the generalization error to identify the stopping criterion. Using the same bias-variance trade-off motivation, having a validation set can aid in devising a stopping criterion, but this has not been feasible in existing zero-shot learning approaches, which either use all acquired measurements (Ulyanov et al., 2018; Senouf et al., 2019) or partition them into two sets for training and defining loss (Hosseini et al., 2020a). Hence, existing zero-shot learning techniques lack a validation set to identify the stopping criterion.

**ZS-SSL Formulation and Training:** We propose a new ZS-SSL partitioning framework to enable subject-specific self-supervised training and validation with a well-defined stopping criterion. We define the following partition for *the available measurement locations from a single scan*, $\Omega$:

$$\Omega = \Theta \sqcup \Lambda \sqcup \Gamma, \tag{7}$$

where $\sqcup$ denotes a disjoint union, i.e. $\Theta$, $\Lambda$ and $\Gamma$ are pairwise disjoint (Figure 1). Similar to Section 2.4, $\Theta$ is used in the DC units of the unrolled network, and $\Lambda$ is used to define a k-space loss for the self-supervision of the network. The third partition $\Gamma$ is a set of acquired k-space indices set aside for defining a k-space validation loss. Thus, ZS-SSL training is both self-supervised and self-validated.

In general, since zero-shot learning approaches perform training using a single dataset, generation of multiple data pairs from this single dataset is necessary to self-supervise the neural network (Quan et al., 2020). Hence, we generate multiple $(\Theta, \Lambda)$ pairs from the acquired locations $\Omega$ of the single scan. In ZS-SSL, this is achieved by fixing the k-space validation partition $\Gamma \subset \Omega$, and performing the retrospective masking on $\Omega\backslash\Gamma$ multiple times. Formally, $\Omega\backslash\Gamma$ is partitioned $K$ times such that

$$\Omega\backslash\Gamma = \Theta_k \sqcup \Lambda_k, \quad k \in \{1, \ldots, K\}, \tag{8}$$

where $\Lambda_k$, $\Theta_k$ and $\Gamma$ are pairwise disjoint, i.e. $\Omega = \Gamma \sqcup \Theta_k \sqcup \Lambda_k, \forall\, k$. ZS-SSL training minimizes

$$\min_{\boldsymbol{\theta}} \frac{1}{K} \sum_{k=1}^{K} \mathcal{L}\Big(\mathbf{y}_{\Lambda_k}, \; \mathbf{E}_{\Lambda_k}\big(f(\mathbf{y}_{\Theta_k}, \mathbf{E}_{\Theta_k}; \boldsymbol{\theta})\big)\Big)$$

In the proposed ZS-SSL, this is supplemented by a k-space self-validation loss, which tests the generalization performance of the trained network on the k-space validation partition $\Gamma$. For the $l^{\text{th}}$ epoch, where the learned network weights are specified by $\boldsymbol{\theta}^{(l)}$, this validation loss is given by:

$$\mathcal{L}\Big(\mathbf{y}_{\Gamma}, \; \mathbf{E}_{\Gamma}\big(f(\mathbf{y}_{\Omega\backslash\Gamma}, \mathbf{E}_{\Omega\backslash\Gamma}; \boldsymbol{\theta}^{(l)})\big)\Big). \tag{9}$$

Note that in (9), the network output is calculated by applying the DC units on $\Omega\backslash\Gamma = \Theta \sqcup \Lambda$, i.e. all acquired points outside of $\Gamma$, to better assess its generalizability performance. Our key motivation is that while the training loss will decrease over epochs, the k-space validation loss will start increasing once overfitting is observed. Thus, we monitor the loss in (9) during training to define an early stopping criterion to avoid overfitting. Let $L$ be the epoch in which training needs to be stopped. Then at inference time, the network output is calculated as $f(\mathbf{y}_{\Omega}, \mathbf{E}_{\Omega}; \boldsymbol{\theta}^{(L)})$, i.e. all acquired points are used to calculate the network output.

**ZS-SSL with Transfer Learning (TL):** While pretrained models are very efficient in reconstructing new unseen measurements from similar MRI scan protocols, their performance degrades significantly when acquisition parameters vary (Muckley et al., 2021). Moreover, retraining a new model on a large database for each acquisition parameter, sampling/contrast/anatomy/acceleration, may be very computationally expensive (Knoll et al., 2019). Hence, TL has been used for re-training DL models pre-trained on large databases to reconstruct MRI data with different characteristics (Knoll et al., 2019). However, such transfer still requires another, often smaller, database for re-training. In contrast, in the presence of pre-trained models, ZS-SSL can be combined with TL, referred to as ZS-SSL-TL, to reconstruct a *single* slice/instance with different characteristics by using weights of the pre-trained model for initialization. Thus, ZS-SSL-TL ensures that the pretrained model is adapted for each patient/subject, while facilitating faster convergence time and reduced reconstruction time.

## 4 EXPERIMENTS

### 4.1 DATASETS

We performed experiments on publicly available fully-sampled multi-coil knee and brain MRI from fastMRI database (Knoll et al., 2020a). Knee and brain MRI datasets contained data from 15 and 16 receiver coils, respectively. Fully-sampled datasets were retrospectively undersampled by keeping 24 lines of autocalibrated signal (ACS) from center of k-space. FastMRI database contains different contrast weightings. For knee MRI, we used coronal proton density (Cor-PD) and coronal proton density with fat suppression (Cor-PDFS), and for brain MRI, axial FLAIR (Ax-FLAIR) and axial T2 (Ax-T2). Different types of datasets and undersampling masks used in this study are provided in Figure 7 in the Appendix.

### 4.2 IMPLEMENTATION DETAILS

All PG-DLR approaches were trained end-to-end using 10 unrolled iterations. CG method and a ResNet structure (Timofte et al., 2017) were employed in the DC and regularizer units of the unrolled network, respectively (Yaman et al., 2020). The ResNet is comprised of a layer of input and output convolution layers, and 15 residual blocks (RB) each containing two convolutional layers, where the first layer is followed by ReLU and the second layer is followed by a constant multiplication (Timofte et al., 2017). All layers had a kernel size of $3 \times 3$, 64 channels. The real and imaginary parts of the complex MR images were concatenated prior to being input to the ResNet as 2-channel images. The unrolled network, which shares parameters across the unrolled iterations had a total of 592,129 trainable parameters. Coil sensitivity maps were generated from the central $24 \times 24$ ACS using ESPIRiT (Uecker et al., 2014). End-to-end training was performed with a normalized $\ell_1$-$\ell_2$ loss (Adam optimizer, LR $= 5 \cdot 10^{-4}$, batch size $= 1$) (Yaman et al., 2020). Peak signal-to-noise ratio (PSNR) and structural similarity index (SSIM) were used for quantitative evaluation.

## 4.3 Reconstruction Method Comparisons

In this work, we focus on comparing training strategies for accelerated MRI reconstruction. Thus, we use the same network architecture from Section 4.2 for all training methods in all experiments. We note that the proposed ZS-SSL strategy is agnostic to the specifics of the neural network architecture. In fact, the number of network parameters is higher than the number of undersampled measurements available on a single slice, i.e. dimension of $\mathbf{y}_\Omega$. As such, different neural networks may be used for the regularizer unit in the unrolled network, but this is not the focus of our study.

**Supervised PG-DLR:** Supervised PG-DLR models for knee and brain MRI were trained on 300 slices from 15 and 30 different subjects, respectively. For each knee and brain contrast weighting, two networks were trained separately using random and uniform masks (Hammernik et al., 2018) at an acceleration rate (R) of 4 (Knoll et al., 2020c). Trained networks were used for comparison and TL purposes. We note that random undersampling results in incoherent artifacts, whereas uniform undersampling leads to coherent artifacts that are harder to remove (Figure 7 in Appendix) (Knoll et al., 2019). Hence, we focus on the more difficult problem of uniform undersampling, while presenting random undersampling results in the Appendix.

**Self-Supervision via Data Undersampling (SSDU) PG-DLR:** SSDU (Yaman et al., 2020) PG-DLR was trained using the same database approach as supervised PG-DLR, with the exception that SSDU performed training only using the undersampled data (Sec. 2.4).

**DIP-Recon:** We employ a DIP-type subject-specific MRI reconstruction that uses all acquired measurements in both DC and defining loss (Senouf et al., 2019; Jafari et al., 2021)

$$\mathcal{L}\Big(\mathbf{y}_\Omega,\ \mathbf{E}_\Omega\big(f(\mathbf{y}_\Omega, \mathbf{E}_\Omega; \boldsymbol{\theta})\big)\Big). \tag{10}$$

We refer to the reconstruction from this training mechanism as DIP-Recon. DIP-Recon-TL refers to combining (10) with TL. As mentioned, DIP-Recon does not have a stopping criterion, hence early stopping was heuristically determined (Figure 8 in the Appendix).

**Parallel Imaging:** We include CG-SENSE, which is a commonly used subject-specific conventional PI method (Pruessmann et al., 1999; 2001), as the clinical baseline quality for comparison purposes.

## 4.4 Automated Stopping and Ablation Study

The stopping criterion for the proposed ZS-SSL was investigated on slices from the knee dataset. The k-space self-validation set $\Gamma$ was selected from the acquired measurements $\Omega$ using a uniformly random selection with $|\Gamma|/|\Omega| = 0.2$. The remaining acquired measurements $\Omega\backslash\Gamma$ were retrospectively partitioned into disjoint 2-tuples multiple times based on uniformly random selection with the ratio $\rho = |\Lambda_k|/|\Omega\backslash\Gamma| = 0.4\ \forall k \in \{1, \ldots, K\}$ (Yaman et al., 2020).

Figure 2a shows representative subject-specific training and validation loss curves at R = 4 for $K \in \{1, 10, 25, 50, 100\}$. As expected, training loss decreases with increasing epochs for all $K$. The k-space validation loss for $K = 1$ decreases without showing a clear breaking point for stopping. For $K > 1$, the validation loss forms an L-curve, and the breaking point of the L-curve is used as the stopping criterion. $K = 10$ is used for the rest of the study, while noting $K = 25, 50$ and $100$

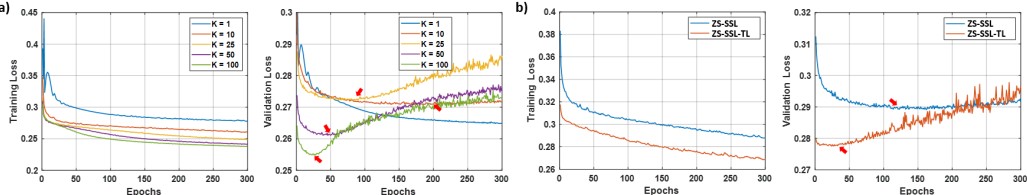

Figure 2: a) Representative training and k-space validation loss curves for ZS-SSL with multiple $K \in \{1, 10, 25, 50, 100\}$ masks on Cor-PD knee MRI using uniform undersampling at R = 4. For $K > 1$ the validation loss forms an L-curve, whose breaking point (red arrows) dictates the automated early stopping criterion for training. b) Loss curves for ZS-SSL with/without TL for $K = 10$ on a Cor-PD knee MRI slice. ZS-SSL with TL converges faster compared to ZS-SSL (red arrows).

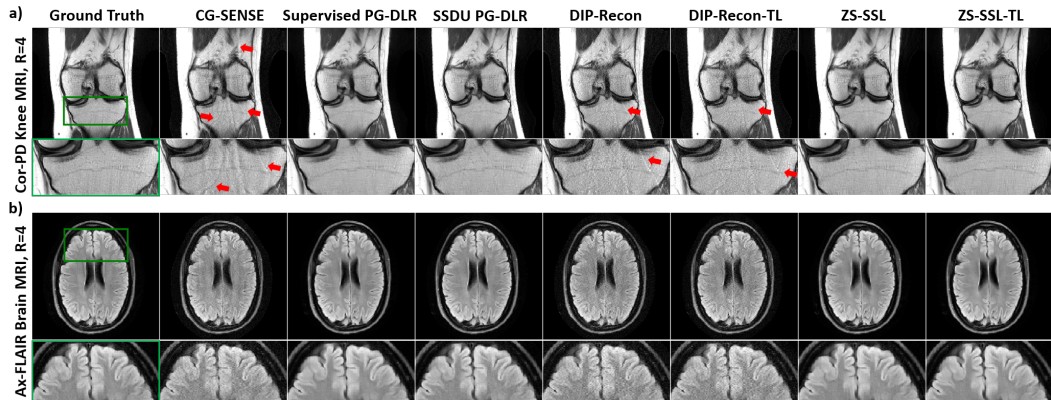

Figure 3: Reconstruction results on a representative test slice from a) Cor-PD knee MRI and b) Ax-FLAIR brain MRI at R = 4 with uniform undersampling. CG-SENSE, DIP-Recon, DIP-Recon-TL suffer from noise amplification and residual artifacts shown with red arrows, especially in knee MRI due to the unfavorable coil geometry. Subject-specific ZS-SSL and ZS-SSL-TL achieve artifact-free and improved reconstruction quality, similar to the database-trained SSDU and supervised PG-DLR.

also show similar performance. Figure 2b shows loss curves on a Cor-PD slice with and without transfer learning. ZS-SSL-TL, which uses pre-trained supervised PG-DLR parameters as initial starting parameters, converges faster in time compared to ZS-SSL, substantially reducing the total training time. Average computation times for single-instance reconstruction methods are presented in Table 2 in the Appendix. Similarly, corresponding reconstruction results for the loss curves in Figure 2a and b are provided in Figure 9 in the Appendix.

## 4.5 RECONSTRUCTION RESULTS

In the first set of experiments, we compare all methods for the case when the testing and training data belong to the same knee/brain MRI contrast weighting with the same acceleration rate and undersampling mask. These experiments aim to show the efficacy of the proposed approach in performing subject-specific MRI reconstruction, while removing residual aliasing artifacts. We also note that this is the most favorable setup for database-trained supervised PG-DLR.

In the subsequent experiments, we focus on the reported generalization and robustness issues with database-trained PG-DLR methods (Knoll et al., 2019; 2020c; Defazio et al., 2020; Muckley et al., 2021). We investigate banding artifacts, as well as in-domain and cross-domain transfer cases. For these experiments, we concentrate on ZS-SSL-TL, since ZS-SSL has no prior domain information, and is inherently not susceptible to such generalizability issues.

**Comparison of Reconstruction Methods:** In these experiments, supervised and SSDU PG-DLR are trained and tested using uniform undersampling at R = 4, representing a perfect match for training and testing conditions. Figure 3a and b show reconstruction results for Cor-PD knee and Ax-FLAIR brain MRI datasets in this setting. CG-SENSE reconstruction suffers from significant residual artifacts and noise amplification in Cor-PD knee and Ax-FLAIR brain MRIs, respectively. Similarly, both DIP-Recon and DIP-Recon-TL suffer from residual artifacts and noise amplification. Supervised PG-DLR achieves artifact-free reconstruction. Both ZS-SSL and ZS-SSL-TL also perform artifact-free reconstruction with similar image quality. Table 1 shows the average SSIM and PSNR values on 30 test slices. Similar observations apply when random undersampling is employed (Figure 10 in the Appendix). For the remaining experiments, we investigate the generalizability of database-pretrained models using supervised PG-DLR as baseline due to its higher performance,

Table 1: Average PSNR and SSIM values on 30 test slices.

|  | Metrics | CG-SENSE | Supervised PG-DLR | SSDU PG-DLR | DIP-Recon | DIP-Recon-TL | ZS-SSL | ZS-SSL-TL |
|---|---|---|---|---|---|---|---|---|
| Cor-PD | SSIM | 0.862 | **0.952** | 0.949 | 0.793 | 0.819 | 0.948 | 0.951 |
|  | PSNR | 34.521 | 39.966 | 39.545 | 32.668 | 33.583 | 39.550 | **40.102** |
| Ax-FLAIR | SSIM | 0.836 | 0.934 | 0.929 | 0.799 | 0.818 | 0.935 | **0.937** |
|  | PSNR | 31.969 | **37.375** | 36.761 | 30.637 | 31.249 | 36.861 | 37.250 |

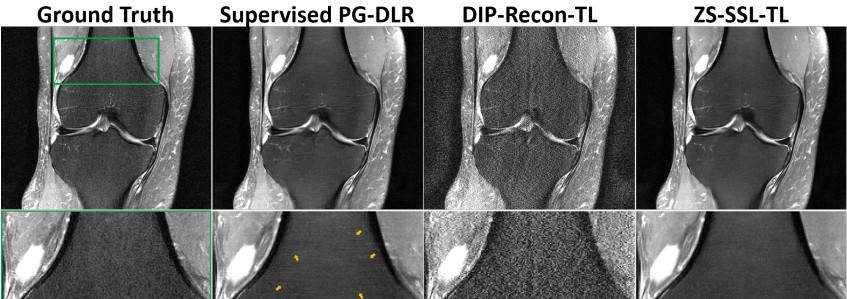

Figure 4: Supervised PG-DLR suffers from banding artifacts (yellow arrows), while ZS-SSL-TL significantly alleviates these artifacts. DIP-Recon-TL suffers from clear noise amplification.

while noting SSDU PG-DLR, which is a self-supervised database-trained model, may also be used as a baseline if needed.

**Banding Artifacts:** Banding artifacts appear in the form of streaking horizontal lines, and occur due to high acceleration rates and anisotropic sampling (Defazio et al., 2020). These hinder radiological evaluation and are regarded as a barrier for the translation of DL reconstruction methods into clinical practice (Defazio et al., 2020). This set of experiments explored training and testing on Cor-PDFS data, where database-trained PG-DLR reconstruction has been reported to show such artifacts (Defazio et al., 2020; Muckley et al., 2021). Figure 4 shows reconstructions for a Cor-PDFS test slice. While DIP-Recon-TL suffers from clearly visible noise amplification, supervised PG-DLR suffers from banding artifacts shown with yellow arrows. ZS-SSL-TL significantly alleviates these banding artifacts in the reconstruction. While supervised PG-DLR achieves slightly better SSIM and PSNR (Table 3 in the Appendix), we note that the banding artifacts do not necessarily correlate with such metrics, and are usually picked up in expert readings (Defazio et al., 2020; Knoll et al., 2020c).

**In-Domain Transfer:** In these experiments, we compared the in-domain generalizability of database-trained PG-DLR and subject-specific PG-DLR. For in-domain transfer, training and test datasets are of the same type of data, but may differ from each other in terms of acceleration and undersampling pattern (Figure 11 in the Appendix). In Figure 5a, supervised PG-DLR was trained with random undersampling and tested on uniform undersampling, both at R = 4. Supervised PG-DLR fails to generalize and suffers from residual aliasing artifacts (red arrows), consistent with previous reports (Knoll et al., 2019; Muckley et al., 2021). Similarly, DIP-Recon-TL suffers from artifacts and noise amplification. Proposed ZS-SSL-TL achieves an artifact-free and improved reconstruction quality. In Figure 5b, supervised PG-DLR was trained with uniform undersampling at R = 4 and tested on uniform undersampling at R = 6. While both supervised PG-DLR and DIP-Recon-TL suffers from aliasing artifacts, ZS-SSL-TL successfully removes these artifacts. Average PSNR and SSIM values align with the observations (Table 3 in the Appendix).

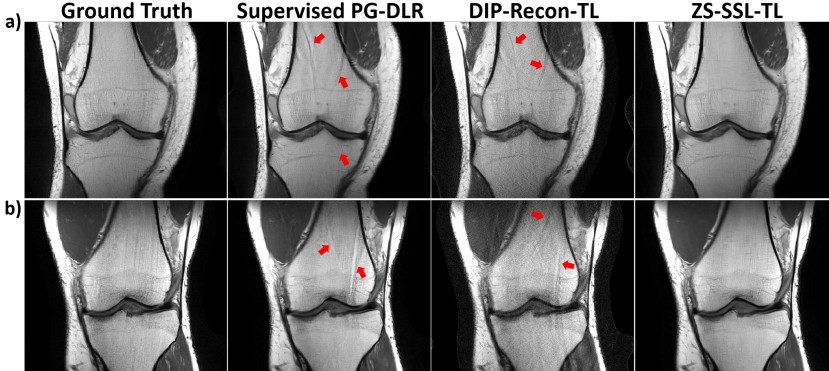

Figure 5: Supervised PG-DLR was trained with a) random mask and tested on uniform mask, both R = 4; b) uniform mask at R = 4 and tested on R = 6 uniform mask. Supervised PG-DLR and DIP-Recon-TL suffer from visible artifacts (red arrows). ZS-SSL-TL yields artifact-free reconstruction.

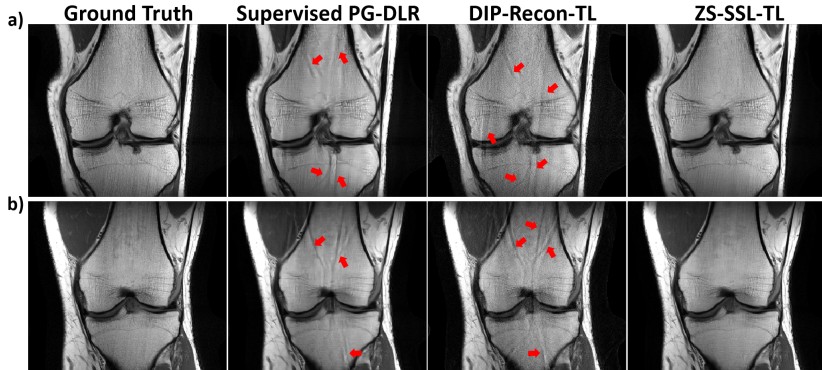

Figure 6: Using pre-trained a) Cor-PDFS (low-SNR) and b) Ax-FLAIR (brain MRI) models for Cor-PD. Supervised PG-DLR fails to generalize for both contrast/SNR and anatomy changes, suffering from residual artifacts (red arrows). DIP-Recon-TL also shows artifacts. ZS-SSL-TL successfully removes noise and artifacts for both cases.

**Cross-Domain Transfer:** In the last set of experiments, we investigated the cross-domain generalizability of database-trained PG-DLR compared to subject-specific trained PG-DLR. For cross-domain transfer, training and test datasets are of the different data characteristics and generally differ in terms of contrast, SNR, and anatomy (Figure 11 in the Appendix). Figure 6 shows results for the case when the testing contrast/SNR and anatomy differs from training contrast/SNR and anatomy, even though the same R = 4 uniform undersampling is used for both training and testing. In Figure 6a, supervised PG-DLR was trained on Cor-PDFS (low-SNR), but tested on Cor-PD (high-SNR and different contrast). In Figure 6b, supervised PG-DLR was trained on Ax-FLAIR (brain MRI) and tested on Cor-PD (knee MRI). In both cases, supervised PG-DLR fails to generalize and has residual artifacts (red arrows). Similarly, DIP-Recon-TL suffers from artifacts and noise. ZS-SSL-TL achieves an artifact-free improved reconstruction. For both cross-domain transfer experiments, similar results were observed for brain MRI (Figure 12 in the Appendix). Average PSNR and SSIM values match these observations (Table 3 in the Appendix)

## 5 CONCLUSIONS

We proposed a zero-shot self-supervised deep learning method, ZS-SSL, for subject-specific accelerated DL MRI reconstruction from a single undersampled dataset. The proposed ZS-SSL partitions the acquired measurements from a single scan into three types of disjoint sets, which are used only in the PG-DLR network, in defining the training loss, and in establishing a validation strategy for early stopping to avoid overfitting. In particular, we showed that with our training methodology and automated stopping criterion, subject-specific zero-shot learning of PG-DLR for MRI can be achieved even when the number of tunable network parameters is higher than the number of available measurements. Finally, we also combined ZS-SSL with transfer learning, in cases where a pre-trained model may be available, for faster convergence time and reduced reconstruction time. Our results showed that ZS-SSL methods perform similarly to database-trained supervised PG-DLR when training and testing data are matched, and they significantly outperform database-trained methods in terms of artifact reduction and generalizability when the training and testing data differ in terms of image characteristics and acquisition parameters. In fact, the subject-specific nature of ZS-SSL ensures that it is agnostic to such changes in acquisition parameters. As such, the proposed work is able to provide good reconstruction quality for each subject, and may have significant implications in the integration of DL reconstruction to clinical studies. We note that hyperparameters, such as learning rate may be adjusted based on the domain for further improvements. It is also noteworthy that the subject-specific ZS-SSL eliminates the requirement for large training sets. This may also facilitate the use and appeal of DL reconstruction for recently developed acquisitions, as well as pilot studies that are often performed to determine the acquisition parameters/acceleration rates of large-scale imaging studies, such as the Human Connectome Project (HCP) (Ugurbil et al., 2013). Finally, while we concentrated on physics-guided models in MRI reconstruction, our ideas and results may inspire further work in related image restoration problems, as well as for generative models or data-driven problems without a data consistency term.

## ACKNOWLEDGEMENT

This work was partially supported by NIH R01HL153146, P41EB027061, U01EB025144; NSF CAREER CCF-1651825. There is no conflict of interest for the authors.

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

# A  APPENDIX

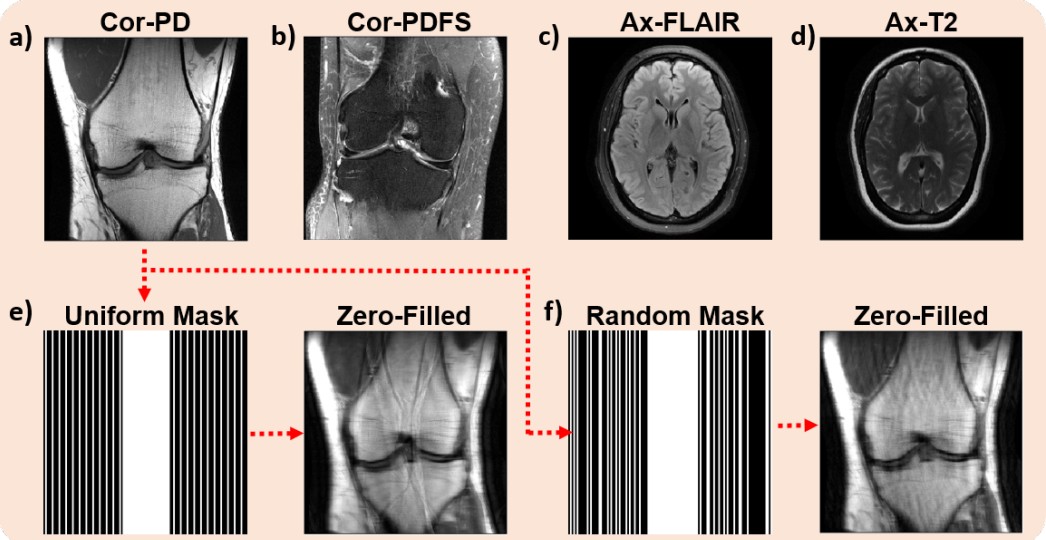

Figure 7: Different contrast weightings and anatomies used in this study: a) Cor-PD, b) Cor-PDFS, c) Ax-FLAIR, d) Ax-T2, as well as undersampling patterns: e) Uniform, f) Random mask. Zero-filled images generated by uniform and random undersampling masks have coherent and incoherent aliasing artifacts, respectively. Coherent aliasing artifacts are generally harder to remove than incoherent artifacts.

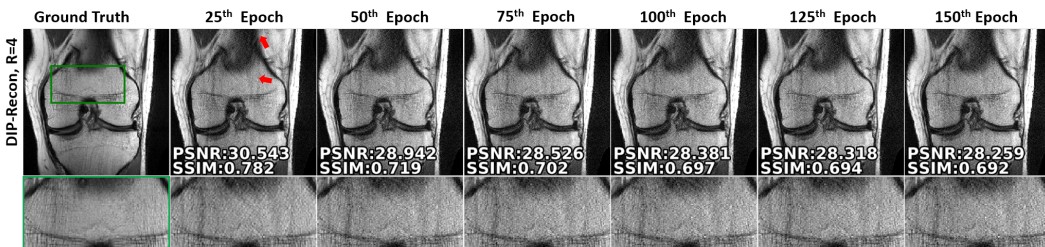

Figure 8: Cor-PD Knee MRI reconstruction results across different epochs for DIP-Recon using uniform undersampling at R = 4. At the 25th epoch, the reconstruction suffers from artifacts, with the zoom-in area showing texture that does not resemble the ground truth. With more epochs, this aspect of the reconstruction improves, but the reconstruction starts to suffer from noise amplification as the number of epochs increases. Hence, the 50th epoch was used in the experiments.

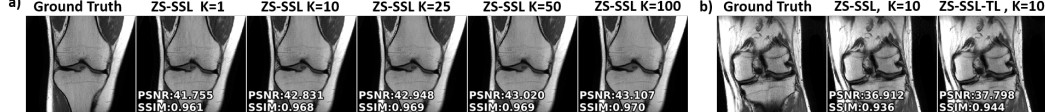

Figure 9: a) and b) show reconstruction results corresponding to the loss curves in Figure 2a and b, respectively.

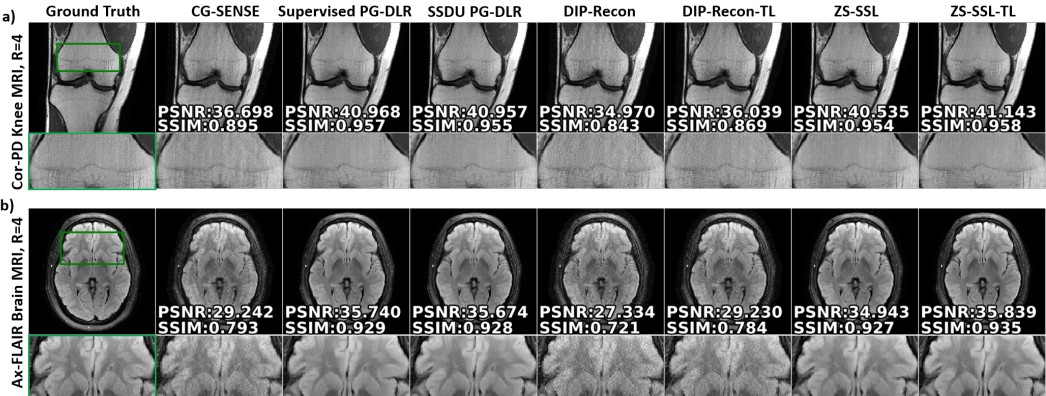

Figure 10: Reconstruction results from R = 4 with random undersampling on representative test slices from a) Cor-PD knee MRI and b) Ax-FLAIR brain MRI. CG-SENSE, DIP-Recon and DIP-Recon-TL suffer from noise amplification. Supervised PG-DLR, SSDU PG-DLR, ZS-SSL and ZS-SSL-TL all show artifact-free reconstruction quality, with similar quantitative metrics.

Table 2: Average reconstruction times for single-instance reconstruction methods. The computation times were measured on the machines equipped with 4 NVIDIA V100 GPUs (each with 32 GB memory). While CG-SENSE and DIP methods have lower computational times, their reconstruction quality is severely degraded hindering clinical usage. ZS-SSL-TL ($K = 10$) provides an 8-fold faster convergence time compared to ZS-SSL ($K = 10$). We note that ZS-SSL methods reconstruction times may further be reduced by means of more compact architectures. Additionally, the increased computational times may be tolerable within the workflow, for instance in clinical settings where image readings are done the next day (Basha et al., 2017), or scans such as high-resolution functional or diffusion MRI, where it is challenging to have high-quality high-resolution data, while post-reconstruction analyses readily take hours to days (Andersson & Stamatios N, 2016; Vizioli et al., 2021).

|  | CG-SENSE | DIP-Recon | DIP-Recon-TL | ZS-SSL | ZS-SSL-TL |
|---|---|---|---|---|---|
| Average Time (sec) | <<1 | 75 | 75 | 640 | 85 |

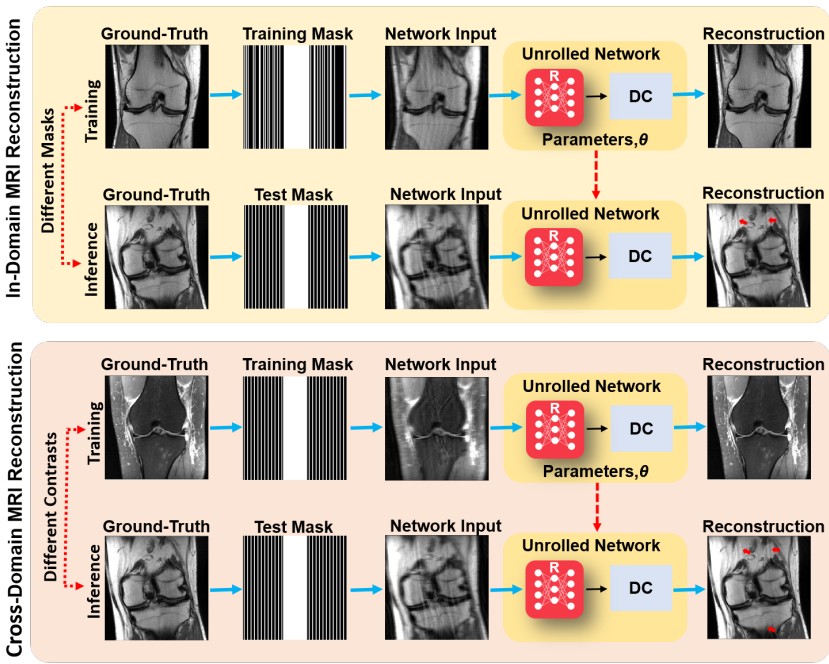

Figure 11: Test datasets may differ from the training datasets in terms of sampling pattern, SNR, contrast and anatomy. Such differences lead to sub-optimal reconstructions in the test datasets, raising robustness and generalizability concerns for translation of trained MRI reconstruction models to clinical practice.

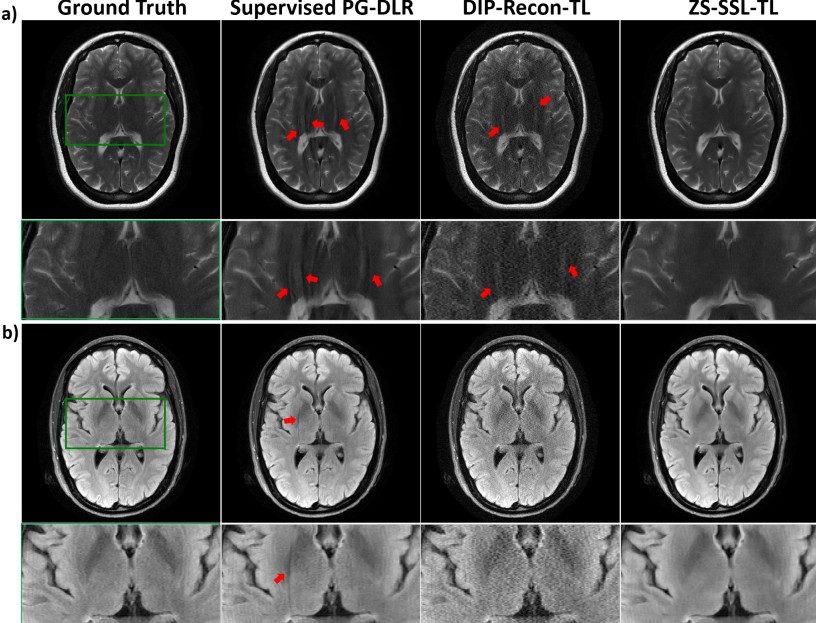

Figure 12: a) Using pre-trained Ax-Flair for Ax-T2 reconstruction. b) Using a pre-trained Cor-PD (knee MRI) for Ax-Flair (brain MRI) reconstructions. Supervised PG-DLR fails to generalize when contrast, SNR and anatomy changes, with residual artifacts (red arrows). DIP-Recon-TL also shows artifacts. ZS-SSL-TL successfully removes noise and artifacts.

Table 3: Average PSNR and SSIM values on 30 test slices for the experiments associated with Figures 4-6 (in the main text) and 12. We note that ZS-SSL-TL successfully fine-tunes the network for each new dataset/instance regardless of the starting pretrained model. Interestingly, there are cases where out-of-domain transfer has slightly higher metrics than in-domain transfer. In these cases, since the metrics are already high, the slight quantitative differences do not affect the overall quality. However, the main difference in these cases is in the convergence/stopping time, where in-domain transfer is typically converges/stops in ∼2-fold fewer iterations than out-of-domain transfer.

| | Metrics | Supervised PG-DLR | DIP-Recon-TL | ZS-SSL-TL |
|---|---|---|---|---|
| Figure 4: Banding Artifacts | SSIM | **0.873** | 0.530 | 0.861 |
| | PSNR | **36.365** | 26.924 | 36.121 |
| Figure 5a: In-Domain Transfer - Different Mask | SSIM | 0.949 | 0.836 | **0.951** |
| | PSNR | 39.167 | 34.093 | **40.088** |
| Figure 5b: In-Domain Transfer - Different Rates | SSIM | 0.937 | 0.792 | **0.940** |
| | PSNR | 38.262 | 32.658 | **38.301** |
| Figure 6a: Cross-Domain Transfer - Knee-Different Contrast | SSIM | 0.931 | 0.859 | **0.949** |
| | PSNR | 37.566 | 34.855 | **39.855** |
| Figure 6b: Anatomy Change - Trained on Brain & Tested on Knee | SSIM | 0.936 | 0.890 | **0.957** |
| | PSNR | 37.494 | 35.458 | **40.407** |
| Figure 12a: Cross-Domain Transfer - Brain-Different Contrast | SSIM | 0.929 | 0.834 | **0.950** |
| | PSNR | 35.578 | 32.655 | **38.767** |
| Figure 12b: Anatomy Change - Trained on Knee & Tested on Brain | SSIM | 0.929 | 0.806 | **0.936** |
| | PSNR | 36.242 | 30.849 | **37.134** |

