# OpenReview forum: "Zero-Shot Self-Supervised Learning for MRI Reconstruction"
_ICLR.cc/2022/Conference — ICLR 2022 Poster_

### Official Review · Reviewer_mBMk · 2021-10-27

**Correctness:** 4
**Technical Novelty And Significance:** 2
**Empirical Novelty And Significance:** 2
**Recommendation:** 5
**Confidence:** 5

**Main Review:**

While the paper is pleasant to read and rigorous it does not offer a significant improvement over ref [Yaman et al., 2020] which by the way is regrettably not included in the set of benchmark methods to which the submission is compared in the result tables of the experimental section.

I am not sure to understand the argument w.r.t. transfer learning and single dataset reconstruction. The proposed approach also retrains the model on a new dataset with patient specific properties. I think that what the authors mean is that this dataset does not need to be supervised. But this is a built-in property of the model from [Yaman et al., 2020] from which the authors are building upon.

The proposed approach as well as supervised PG-DLR seem to fail to reconstruct texture feature visible in ground truth images (the reconstructed ones are smoother). This might be a problem for diagnosis or for other trained ML based models that rely on texture features such as radiomics and take MR images as inputs. Is this a known issue in MRI reconstruction in general or in DL based reconstruction ?

**Summary Of The Paper:**

This submission deals with MR images reconstruction in a context where the raw data is under-sampled. This data (which is in the Fourier domain) can be under-sampled to accelerate the imaging exam and thus improve clinical workflow and it is thus a very relevant research topic.

The submission relies on recent deep learning models that explicitly incorporate some physical aspects of MR image acquisition (multiple coils, coil sensitivity and Fourier transform) and achieve MRI reconstruction based on supervised training examples. The training examples do not need to be (under-sampled / fully sampled) pairs. Indeed, some entries of under-sampled data can be deleted to create an input for which correct reconstruction of the deleted entries can be required through a computable loss term.


**Summary Of The Review:**

Pros :
- the paper is well written and ideas are clearly explained and stated.
- the paper is technically sound.

Cons :
- the contribution is too incremental compared to [Yaman et al., 2020] as it consists in re-using the Fourier domain partitioning idea in order to have a validation loss allowing to detect overfitting and stop learning.
- patient specific training demands more computation time and resources than usual clinical workflow.

---

> ### Author Response · Authors · 2021-11-17
> **Initial Point by Point Rebuttal (and invitation for further discussion) - Part 1**
>
> #### **mBMk.C1:  Major differences compared to [Yaman et al. 2020]**
>
> We thank the reviewer for this valuable comment. As the reviewer suggested we have now included the comparisons with [Yaman et al. 2020] in Figure 3 (and Figure 10 in the Appendix), as well as in Table 1. We note that both supervised PG-DLR and [Yaman et al. 2020] are database training approaches, which are susceptible to the same set of generalization issues. These new results show supervised PG-DLR slightly outperforms [Yaman et al. 2020], consistent with what was reported in [Yaman et al. 2020]. Therefore, since both methods are database-trained, for the remainder of the experiments that check the generalizability of pretrained models, we used pretrained supervised PG-DLR. This is now clarified in the text as well.
>
>
>
> Furthermore, while our work share similarities with [Yaman et al, 2020], it has major differences:
>
>
>
> *Database vs. zero-shot learning:*  [Yaman et al, 2020] is a self-supervised database deep learning approach and thus requires a large database of undersampled measurements to perform training,whereas the proposed ZS-SSL is a subject-specific/single-instance deep learning reconstruction approach, and does not require any external database. As detailed in the text, a mismatch between training and test data in terms of sampling/contrast/anatomy /acceleration rate/vendor leads to generalizability issues for database-trained methods that result in reconstruction failure [1]. In contrast, the proposed ZS-SSL performs training and testing on the same single subject/instance, and is inherently not susceptible to such issues. Hence, zero-shot subject-specific learning is desirable to avoid potential misdiagnosis due to such artifacts.
>
>
> *Novel partitioning for automated stopping criterion in zero-shot learning:* A naive application of the partitioning scheme in [Yaman et al, 2020] to the zero-shot setting fails due to overfitting, since there is no stopping criterion [2]. Thus, an automated stopping criterion for zero-shot learning with broad applicability is critically needed. Our proposed framework is well-aligned with the use of a validation set in the conventional database-trained setting, where the bias-variance trade-off is used to motivate early stopping as regularization, and the stopping criterion is identified using a separate validation set.  Using the same bias-variance trade-off motivation, having a validation set can help us devise a stopping criterion, but this has not been feasible in other zero-shot learning approaches due to the partitioning framework they have used.
>
> Therefore, unlike existing self-supervised models in the zero-shot setting, our proposed ZS-SSL partitions measurements into three sets, where two sets are used for self-supervision and the last set is used for self-validation, where the validation set serves as a proxy for generalization error that balances the bias-variance trade-off with the proper stopping criterion. Hence, ZS-SSL tackles the challenge of the lack of validation set in zero-shot learning for MRI reconstruction, which may inspire further investigations in zero-shot learning applications in other domains. While it may seem straightforward in hindsight, we think that this is a major innovation. In a way, this is similar to how holdout masking (or the split into two sets) seems natural for self-supervision now, but was entirely novel when first proposed.
>
>
> *Medical imaging applications and impact:* As detailed in the previous points, our work solves a major problem in medical imaging reconstruction. A database-trained model (e.g. [Yaman et al. 2020] or supervised training) is prone to failure when applied to patients/scans with different characteristics, with potential for misdiagnosis [1]. ZS-SSL avoids this risk by adapting the model to each individual without overfitting. Hence, we believe it can play an essential role for integration of deep learning reconstruction to clinical studies, and such medical applications are practically very impactful.
>
>
>
> Furthermore, zero-shot learning is important in certain clinical and practical applications, where it is difficult to procure large training databases, such as whole 3D/4D volumes of functional, diffusion or perfusion MRI. This is especially important for translational studies that propose new acquisition schemes at ultra-high resolutions. However, collecting 300 volumetric samples for database training as in [Yaman et al, 2020], at a pace of one subject per day would take nearly a year!  Acquisition of such a database would be time-consuming, costly and impractical, especially for techniques that are in development and/or higher resolution than clinical settings. However, it is precisely these techniques, where deep learning reconstruction may be very beneficial. Thus, a zero-shot learning approach will greatly help a large number of MRI studies focusing on technique development, prior to their clinical translation .

---

> > ### Author Response · Authors · 2021-11-17
> > **Initial Point by Point Rebuttal (and invitation for further discussion) - Part 2**
> >
> > #### **mBMk.C2:  Argument w.r.t. transfer learning and single dataset reconstruction.**
> >
> > We apologize for the confusion. It seems that this confusion stems from the fact that we used the word "dataset" to refer to a single slice/instance, while we reserved the word "database'' to describe multiple such slices/instances. To clarify, proposed ZS-SSL performs training and testing only on a single slice/instance, without any external training database or pretrained models. However, its computation time can be lengthy, as it uses random weights for initialization. Therefore, in the case when a model pretrained on a database is available,  ZS-SSL can be further combined with transfer learning to reduce computational costs, referred to as ZS-SSL-TL in our paper. Thus, when ZS-SSL is combined with the TL, it does not retrain a model on a new database, but only on that single slice/instance by initializing the weights of the model with the pretrained model for computational gains.
> >
> > Furthermore, since we impose no restrictions on the database-pretrained model, it may be pretrained using the approach of [Yaman et al.]. This would still require a database containing hundreds of undersampled measurements/slices, although unlike supervised training, it does not require the database to be fully-sampled. Therefore, ZS-SSL-TL can also use a pretrained model from [Yaman et al. 2020] for initialization if needed. Related changes are now incorporated into the text.
> >
> >
> > #### **mBMk.C3:  Smoothness in DL reconstructions.**
> >
> > Thank you for this excellent observation. Indeed, deep learning based MRI reconstruction faces smoothness issues, especially at high accelerations. This is an ongoing problem that was pointed out in the fastMRI challenge [1]. A reader study conducted with radiologists regard the R=4 deep learning reconstructions as acceptable, but higher rates such as R=8 are often deemed as not acceptable from a clinical perspective due to smoothness/blurring. Another challenge that can be regarded as a problem for diagnois is the banding artifacts that was highlighted in the earlier fastMRI challenge [3]. As our results show that, ZS-SSL significantly alleviates these artifacts which can facilitate the use of deep learning in clinical studies.
> >
> >
> > #### **mBMk. Summary of Review- Incremental contribution compared to [Yaman et al. 2020]**
> > We hope that we readily explained the differences between [Yaman et al, 2020] and our work in detail. As discussed, in a clinical setting, it is imperative to provide high-quality reconstructions that can be used to identify lesions/disease for every individual. We and others have shown that database-trained deep learning MRI reconstruction faces generalization issues, which may lead to misdiagnosis and potentially to severe outcomes. In contrast, the proposed zero-shot patient-specific learning approach ensures high-quality reconstructions for every individual/patient. Furthermore, we also highlighted that in certain translational applications, it may be infeasible to procure large training databases for database training methods such as [Yaman et al, 2020]. This further highlights the importance and need for zero-shot learning in MRI reconstruction. Thus, we hope that the reviewer will appreciate the importance and timeliness of our work for pressing problems in MRI reconstruction and health care.
> >
> >
> > #### **mBMk. Summary of Review- patient specific training demands more computation time**
> > As the reviewer suggests patient-specific training demands more computation time at inference compared to pretrained models. However, the additional computational time may be acceptable in certain applications since ZS-SSL resolves several issues associated with pretrained models, such as generalizability and banding artifacts that hinder the use of deep learning reconstruction in clinical studies.
> > Additionally, we have now provided the computation times for all single instance reconstruction approaches in the text, and also noted that these can be further reduced with more compact networks. While zero-shot learning computation times can be improved with changes in the architecture, for some applications the computation times may be tolerable within the workflow. Such applications include cases when image readings are done the next day in clinical workflow [4], or scans such as functional or diffusion MRI where it is challenging to have high-quality high-resolution data as discussed earlier, while post-reconstruction analyses take hours to days regardless [5,6].
> >
> >
> >
> > Finally, we hope that we clarified these very helpful comments. We would be glad to discuss further.
> >
> > References:
> > [Yaman et al, 2020] DOI: 10.1002/mrm.28378.
> >
> > [1] DOI: 10.1109/TMI.2021.3075856.
> > [2] DOI: 10.1109/EMBC44109.2020.9176241.
> > [3] DOI: 10.1002/mrm.28338.
> > [4] DOI: 10.1002/jmri.25695.
> > [5] DOI: 10.1016/j.neuroimage.2015.10.019.
> > [6] DOI:10.1038/s41467-021-25431-8.

---

> > > ### Comment · Reviewer_mBMk · 2021-11-19
> > > **one more question**
> > >
> > > I thank the authors for their detailed answers. I do not have more observations.
> > >
> > > Just one more question concerning computation times reported in Table 2 of the appendices : Can theses results be contrasted with the time that is saved by using a fast-MRI compared to a full resolution MRI exam ?

---

> > > > ### Author Response · Authors · 2021-11-19
> > > > **Computational times vs. savings in acquisition times**
> > > >
> > > > We thank the reviewer again for their continued efforts and interest in improving our work. We are also glad to hear that we have answered the reviewer’s comments.
> > > >
> > > > Admittedly, this new answer is a bit tricky to put in context. First off, since we used the fastMRI database for the data, we do not know the exact acquisition times. But it is safe to say that the per slice acquisition time with/without acceleration is going to be lower than the subject-specific reconstruction times of all DL methods. However, in most applications, the time savings from accelerated imaging are usually considered to be decoupled from the computational time (except in cases that require real-time feedback like interventional imaging).
> > > >
> > > > To expand on this, we want to highlight that accelerated/fast MRI is used in two different context:
> > > >
> > > > 1) It is used to reduce the overall scan time. For instance, for the anatomical data presented here, this leads to shorter overall exam times. In turn, this reduces patient discomfort, costs and the likelihood of motion artifacts. These benefits are usually considered more important in the clinic than lengthier reconstruction times, especially if the images do not need to be evaluated immediately (e.g. [1,2,3]) as discussed before.
> > > >
> > > > 2) In some exams where there is a fixed or minimal time cost due to physical imaging constraints (e.g. acquiring an image of the heart every heartbeat with different contrast uptake for perfusion MRI or ensuring signal regrowth in diffusion MRI), then it is used to not reduce the scan time but to improve the resolution/coverage to image structures not detectable with current technologies. In this case, the overall scan time would remain the same, but the information content would improve. Again, the computational costs may be tolerable in this setting, since higher-quality data may be more important than longer reconstruction & analysis times.
> > > >
> > > > In replying to this comment, we also realized that we had not included the discussion on when higher computational costs may be tolerable in the text during our first revision, hence we have added these now to the Table 2 caption and uploaded the modified PDF.
> > > >
> > > > Thank you again for pointing out these subtle distinctions. Please let us know if we can clarify further.
> > > >
> > > > Finally, we would be glad to engage in discussion if there are any other questions
> > > >
> > > > References
> > > >
> > > > [1] DOI: 10.1002/jmri.25695.
> > > > [2] DOI: 10.1016/j.neuroimage.2015.10.019.
> > > > [3] DOI:10.1038/s41467-021-25431-8.

---

### Official Review · Reviewer_19v3 · 2021-11-05

**Correctness:** 3
**Technical Novelty And Significance:** 2
**Empirical Novelty And Significance:** 3
**Recommendation:** 5
**Confidence:** 4

**Details Of Ethics Concerns:**

No concern

**Main Review:**

The paper is well written and clear with great illustrations and only a few typos. It applies the combined principles of zero-shot learning [1] plug-and-play [2] iterative architectures and transfer learning to solve the MRI reconstruction problem on single images.

The idea of zero-shot learning is not new, in the particular context of imaging, it has long been known that the denoising problem could be solved by training on the very same image one would seek to denoise [3,4]. Plug-and-play architectures combine a general-purpose denoiser with a Tikhonov-like least-square solver to solve arbitrary inverse problems. MRI is often regarded as a classical inverse problem in the same range as CT reconstruction or deblurring. The range of solutions provided to solving inverse problems in medical imaging is too large to mention [5].

From the methodological point of view, I see very little novelty in this article.

There are many aspects to using zero-shot learned approaches for denoising, in particular the fact that their regularity is not established and so they can only be applied for a limited number of iterations, and the need to use early-stopping. One contribution of this article is their proposed condition for early stopping, but it is not studied mathematically.





[1] Romera-Paredes B, Torr P. An embarrassingly simple approach to zero-shot learning. InInternational conference on machine learning 2015 Jun 1 (pp. 2152-2161). PMLR.
[2] S. Venkatakrishnan, C. A. Bouman and B. Wohlberg, "Plug-and-play priors for model based reconstruction", IEEE Global Conference on Signal and Information Processing, pp. 945-948, 2013.
[3] Huang DA, Kang LW, Wang YC, Lin CW. Self-learning based image decomposition with applications to single image denoising. IEEE Transactions on multimedia. 2013 Oct 7;16(1):83-93.
[4] Liu C, Szeliski R, Kang SB, Zitnick CL, Freeman WT. Automatic estimation and removal of noise from a single image. IEEE transactions on pattern analysis and machine intelligence. 2007 Dec 18;30(2):299-314.
[5] Senouf O, Vedula S, Weiss T, Bronstein A, Michailovich O, Zibulevsky M. Self-supervised learning of inverse problem solvers in medical imaging. InDomain adaptation and representation transfer and medical image learning with less labels and imperfect data 2019 Oct 13 (pp. 111-119). Springer, Cham.

**Summary Of The Paper:**

This article propose a "zero-shot" method for MRI reconstruction, which is a well-studied inverse problem. The method is based on the ideas of deep image prior, i.e. the ability of correctly sized neural networks to learn about the structure of a single image, sufficiently well for denoising tasks. This self-supervised learned network can then be used in a plug-and-play architecture to solve inverse problem with a variational approach, i.e. as if the learned denoiser were a Total Variation (TV) minimiser. Their denoiser can be improved in a transfer learning approach to benefit from a more complex network trained on similar images than those at hand, and fined-tuned on the image to be reconstructed

The authors go on to apply their plug-and-play architecture to solve the MRI reconstruction problem. They provide experiments and comparisons.

**Summary Of The Review:**

The authors make a big deal of using TL, 0-shot and plug-and-play to achieve good results on MRI reconstruction. All of these elements are known and have been applied before to this problem. They do achieve better results than Senouf et al [5], thanks to a better TL regimen.

The paper is correct overall. The early stopping condition is debatable and not studied theoretically.

Overall the paper is pretty good, but I think it does not clear the bar for acceptance at ICLR due to the lack of technical novelty.

---

> ### Author Response · Authors · 2021-11-17
> **Initial Point by Point Rebuttal (and invitation for further discussion)**
>
> #### **19v3.Comment: On technical novelty**
>
> We agree that zero-shot learning approaches in denoising domain requires early stopping, since they perform training and testing only on a single image. Hence, there are no samples/validation set that can be used as a proxy for generalization error. This is unlike conventional database-trained setting, where the bias-variance trade-off is used to motivate early stopping as regularization, and the stopping criterion is identified using a separate validation set.  Using the same bias-variance trade-off motivation, a validation set will help us devise a stopping criterion, but this has not been feasible in other zero-shot learning approaches due to the partitioning framework used. Unlike existing self-supervised models in zero-shot setting, proposed ZS-SSL partitions measurements into three sets, where two sets are used for self-supervision and the last set is used for self-validation, serving as a proxy for generalization error that balances the bias-variance trade-off with the proper stopping criterion. Hence, ZS-SSL tackles the challenge of the lack of validation set in zero-shot learning for MRI reconstruction, which may inspire further investigations in zero-shot applications in other domains. Upon reviewer's suggestion, we have now modified the text to further clarify the importance of the proposed ZS-SSL.
>
>
> We also kindly note that while it may seem straightforward in hindsight, we think that our partitioning framework for joint self-supervision and self-validation is a major innovation. In a way, this is similar to how holdout masking (or the partition into two sets) seems natural for self-supervision now, but it was entirely novel when first proposed.
>
>
> Finally, we want to highlight the importance of our work for medical imaging, which we believe fully aligns with the ICLR application topic on health care. A database-trained model (e.g. supervised training) is prone to failure when applied to patients/scans with different characteristics, with potential for misdiagnosis [1]. ZS-SSL avoids this risk by adapting the model to each individual without overfitting. Hence, we believe it can play an essential role for integration of deep learning reconstruction to clinical studies, and such medical applications are practically very impactful.
>
>
>
> #### **19v3.Summary: Better results than Senouf et al, thanks to better TL regimen. Lack of details on early stopping motivation and theory.**
>
> We kindly note that the proposed approach does not depend on TL. Our ZS-SSL enables subject-specific MRI reconstruction without any external dataset, providing a well-defined stopping criterion that avoids overfitting. However, we also noted that it can be further combined with TL to reduce computational costs, in the case that pretrained models are available, referred to as ZS-SSL-TL in our paper. Specifically, we can also use pretrained models with a mismatch between training and test data characteristics as a starting point for ZS-SSL-TL, though it is well-known that such models will suffer from generalizability issues at testing [1]. Thus, the use of TL is for faster convergence time, so that such techniques can be integrated in clinical settings. Otherwise, ZS-SSL does not require a TL regimen, and can be initialized from random weights, with excellent performance, as shown in Figure 3 and Table 1.
>
>
> We also note that Senouf et al is a database learning method that uses all acquired measurements for training and defining loss. Its reconstruction performance has been shown to suffer from noise amplification [1]. Its subject-specific version would be the DIP-Recon presented here, which similarly suffers from major noise amplification and artifacts. In contrast, ZS-SSL alleviates such artifacts.
>
> Lastly, we now revised the text and used the bias-variance trade-off to motivate how the validation set is used to establish stopping criterion. While we hope this provides more intuition about why and how we devised our early stopping criterion, we do acknowledge that it is a high-level description. We agree with the reviewer that a detailed theoretical analysis is desirable, but image reconstruction is a notably difficult area for such analysis [2], where even the theoretical underpinnings of database-trained self-supervised training are not fully understood, in contrast to image denoising [3]. We hope that our application-focused paper will generate interest and be applied to other problems, such as denoising, and a detailed analysis may perhaps be devised eventually. However, given the scope and timeliness of our work for a pressing problem in clinical MRI, we will appreciate if our paper can be evaluated in the context of ICLR application topics, specifically biomedical imaging and health care.
>
> We hope that we clarified these helpful comments. We would be glad to discuss further.
>
> [1] DOI:10.1109/TMI.2021.3075856, 2021.
> [2] arXiv:2105.08040, 2021.
> [3] arXiv: 1901.11365, 2019.

---

### Official Review · Reviewer_pt6r · 2021-11-06

**Correctness:** 3
**Technical Novelty And Significance:** 3
**Empirical Novelty And Significance:** 3
**Recommendation:** 6
**Confidence:** 4

**Main Review:**

####################################################

Strengths

+ Overall, this is a very interesting paper with impressive experimental results. The proposed algorithm yields high-quality reconstructions and the rationale behind designing such an algorithm is very well-motivated by the authors. Indeed, it is important to come up with reconstruction algorithms that do not require large training sets, especially in medical imaging where creating datasets is a tedious task.

+ The idea of self validation is also interesting since it mimics the presence of a validation set for training and prevents the network from overfitting.

+ The robustness investigations are appealing and very related to the problem considered. Moreover, from the robustness perspective, any effort toward single-instance reconstruction is of great value provided that it results in more robustness. Regarding robustness evaluations, please see the reviewer’s comment below.

################################################

Weaknesses/comments

- The paper states several times that ZS-SSL-TL significantly reduces convergence time. Yet what is missing is a computational comparison among single-instance reconstruction methods such as traditional sparsity-based, DIP-based, ZS-SSL, ZS-SSL-TL. This is important since one of the critical challenges of single-instance reconstruction algorithms is their inefficiency at the inference. Adding such an experiment would also make the paper even stronger.

- Table 2 in the appendix contains very interesting results. However, there are several surprising scores achieved by ZS-SSL-TL. For instance, how come for ``trained on brain & tested on knee`` ZS-SSL-TL achieves 40.4 dBs, yet according to Table 1, ``trained on knee & tested on knee`` achieves 40.1 dBs? This is counter-intuitive in that it suggests pre-training a model on ``brains`` is marginally better than pre-training it on ``knees`` if one wants to get a good PSNR on ``knees``!

- Page 7, Section 4.5, Paragraph 2, last sentence: ``ZS-SSL has no prior domain information and is inherently not susceptible to such generalizability issues.`` Is there any evidence to suggest that this is the case? For example, DIP-based methods have been suggested to be susceptible to distribution shifts although there’s no ``prior domain information`` involved. But their hyper-parameters are tuned on a specific domain. Doesn’t ZS-SSL rely on any tunable hyper-parameters that differ from one domain to another?

- The reviewer perceives the point made in Figure 2 regarding automatic early stopping, however, there are the following two comments in terms of the consistency of the results:

    - Second plot from the left: Can the authors provide intuition on why the curves are uncorrelated w.r.t K? Specifically, at the convergence (epoch 300), the curve for k=10 is on top of k=1, the curve for k=25 is on top of k=10, but then somehow the curve for k=50 is below k=25. Moreover, the breakout point of k=50 has a sudden drop from the breakout point of k=25; does this mean k=100 would go down even further?

    - The right-most plot: The reviewer is unsure what the authors mean by ``ZS-SSL with TL converges faster compared to ZS-SSL`` in the caption, in that the validation loss implies no benefit comes with combining ZS-SSL and TL and it’s not a matter of convergence.

- Table 1 suggests that DIP-TL performs strangely poorly compared to PG-DLR and ZS-SSL-TL. Does this mean that the pretrained network used for DIP-TL performs poorly and DIP has not been able to improve its performance, and thus the low score has nothing to do with the DIP itself?


**Summary Of The Paper:**

In order to eliminate the need for large training sets, one can consider a transition from (1) fully-supervised to (2) self-supervised methods, and then from (2) self-supervised methods to (3) single-instance reconstruction methods.

In the context of accelerated MRI reconstruction which is considered in this work, the above categories translate into models that are trained based on (1) having access to a fully-sampled dataset, (2) having access to a dataset of under-sampled measurements, and (3) having access to only one under-sampled measurement. The paper targets (3), that is proposing a zero-shot learning approach for accelerated MRI reconstruction.

The algorithm is based on the idea proposed in paper [1] combined with building a dataset for the given sample in order to eventually perform self-supervised training on the synthesized dataset. In order to prevent overfitting, the authors propose a way to do automatic early stopping.

[1] Yaman, Burhaneddin, et al. "Self-supervised physics-based deep learning MRI reconstruction without fully-sampled data." 2020 IEEE 17th International Symposium on Biomedical Imaging (ISBI). IEEE, 2020.

**Summary Of The Review:**

Decision: Accept (6)

The reviewer finds the paper of great interest to the community and the thorough experimental analysis of the proposed algorithm is the main motivation for accepting the paper. However,

(1) several concerns/comments mentioned above, and

(2) the fact that the major difference between the proposed algorithm and the prior work [1] is the self-validation step combined with dataset synthesis

prevent the reviewer from giving a higher score.

[1] Yaman, Burhaneddin, et al. "Self-supervised physics-based deep learning MRI reconstruction without fully-sampled data." 2020 IEEE 17th International Symposium on Biomedical Imaging (ISBI). IEEE, 2020.

---

> ### Author Response · Authors · 2021-11-17
> **Initial Point by Point Rebuttal (and invitation for further discussion) - Part 1**
>
> #### **pt6r.C1:  The computational comparisons...**
>
> We thank the reviewer for this valuable comment. Indeed, it was our oversight to not provide the quantitative computational improvement of ZS-SSL-TL over ZS-SSL in the original submission. We have now added the average computation time of all reconstruction methods in Table 2. We see that ZS-SSL-TL (*K=10*) provides an 8-fold faster computational time compared to ZS-SSL. Also noteworthy is that CG-SENSE and DIP (epoch number is empirically set to 50 as described in Sec. 4.3) have lower run times, which is due to the lack of multiple masks used in ZS-SSL. Yet, these approaches also suffer major noise amplification and artifacts that would hinder their use in clinical studies. It may be possible to further reduce the reconstruction times of ZS-SSL further by means of more compact architectures in terms of the number of unrolls or depth of neural networks, though this may also impact performance slightly. We have now incorporated these changes into the text.
>
>
> In relation to the computational costs, we also note that zero-shot learning is important in certain MRI applications, where it is difficult to procure large training databases. For instance, there is increasing interest in using PG-DLR with 3D/4D networks to process whole volumes of functional, diffusion, and/or perfusion MRI. This is especially important for translational/pre-clinical studies that propose new acquisition schemes at ultra-high resolutions. However, collecting 300 volumetric samples at a pace of one subject per day would take nearly a year! This would be time-consuming, costly and impractical, especially for techniques that are in development and/or higher resolution than clinical settings. However, it is precisely these techniques, where deep learning reconstruction may be very beneficial. The proposed ZS-SSL would greatly help a large number of MRI studies focusing on technique development, prior to their clinical translation. Similarly, for these applications, the longer computation times of ZS-SSL may not be problematic, since post-reconstruction analyses readily takes hours to days [1,2]. Another potential setting where longer computation times are acceptable would be cases, where image readings are done hours or a day after the acquisition in the clinical workflow [3].
>
> #### **pt6r.C2:  Table 2 results...**
> Thank you for pointing out this subtle observation. The reconstruction quality in both cases align with our expectation that the ZS-SSL-TL will fine-tune the network for each dataset/slice regardless of the pretrained model. Hence, we do expect the reconstruction quality to be similar regardless of the starting pre-trained model, but there may be slight quantitative difference. However, as the reviewer points out, we do expect the reconstruction to be ``easier'' when using the model pretrained on the knee and testing on the knee compared to pretraining on the brain and testing on the knee. But given the excellent reconstruction quality for both (over 40dB PSNR), this difference now mainly lies in the convergence/stopping time. In fact, when we check the convergence time, model pretrained \& tested on the knee is stopped on average in 23 epochs, whereas pretrained on the brain \& tested on the knee is stopped on average in 44 epochs. Thus, while there is almost a ~2-fold reduction in computation time for in-domain transfer, we see similar metrics for both cases at the corresponding epochs where stopping is applied. We note the slight difference in metrics can go either way since the reconstruction quality and metrics are already high at these points. We have now incorporated this explanation into the table caption.
>
>
> #### **pt6r.C3: ZS-SSL has no prior domain information?**
>
> Supervised PG-DLR performs training for a certain set of acquisition-dependent parameters, such as sampling rate or image contrast. Hence, testing performance of supervised PG-DLR heavily depends on aligning with these parameters used during training. Any changes between training and testing may lead to generalizability issues. On the other hand, ZS-SSL is agnostic to such prior information, as it performs training and testing directly on the same acquisition-dependent parameters. Hence, there will not be any in-domain or out-of-domain shifts between training and testing. Therefore, to show the comparison between generalizability, we use ZS-SSL-TL which uses pretrained models as an initialization, and then performs fine-tuning on the characteristics of the scan of interest. However, the reviewer is again right about the nature of hyperparameters, such as the learning rate, which may differ for ZS-SSL based on the domain. In our experiments, we did not tune such hyperparameters for consistency across domains, and still ended up with good experimental results. Optimization of these hyperparameters may further improve the results, though this may be challenging in a practical setting .

---

> > ### Author Response · Authors · 2021-11-17
> > **Initial Point by Point Rebuttal (and invitation for further discussion) - Part  2**
> >
> > ####  **pt6r.C4:  Regarding the points made in Figure 2**
> >
> > *Part1-Regarding K value:*  As *K* increases, the single scan data gets augmented further. Thus, the network learns the parameters in fewer epochs. Hence, as K increases, we see more steep decays in the loss, leading generally to lower validation errors. Therefore, as the reviewer anticipated the *K* = 100 would go down even further. We have now also included *K*=100 in our loss curves to further confirm the reviewer's hypothesis.
> >
> > *Part2-The right-most plot:*  We agree that it is not a matter of convergence, rather a convergence/stopping time. We have now corrected our statements accordingly.
> >
> >
> >
> >
> > #### **pt6r.C5:  Table1 metrics-DIP-Recon-TL**
> >
> >
> > We used the same pretrained network for ZS-SSL-TL and DIP-Recon-TL. However, as discussed in the text, DIP has no stopping criterion, and its stopping epoch was chosen empirically. Even though we start from the same pretrained supervised PG-DLR, DIP-recon-TL would start to update parameters across these epochs until stopped. Noting that DIP-type reconstructions use all acquired measurements $\Omega$ for both training and defining the loss, and since $\Omega$ is used in enforcing data consistency in the network, using $\Omega$ also for defining the loss  essentially amounts to calculating the loss over residual artifacts \& noise. As we have shown in DIP-Recon, such usage of $\Omega$ leads to noise amplification and artifacts in the reconstructions. Therefore, DIP-Recon-TL also suffers from noise amplification, and thus its metrics are lower than PG-DLR and ZS-SSL-TL.
> >
> >
> >
> > #### **pt6r.Summary- Difference between ZS-SSL and [Yaman et al., 2020]**
> >
> > We would like to further clarify the major differences of our proposed work and [Yaman et al,2020], as well as its novelty:
> >
> >
> > *Database vs. zero-shot learning:* [Yaman et al, 2020] is a database self-supervised deep learning  whereas the  proposed ZS-SSL is a subject-specific/single-instance deep learning reconstruction approach, and does not require any external database. As detailed in the text, a mismatch between training and test data in terms of sampling/contrast/anatomy /acceleration rate/vendor leads to generalizability issues for database-trained methods that result in reconstruction failure [4]. In contrast, the proposed ZS-SSL performs training and testing on the same single subject/instance, and is inherently not susceptible to such issues. Hence, zero-shot subject-specific learning is desirable to avoid potential misdiagnosis due to such artifacts.
> >
> >
> >
> >
> >
> > *Novel partitioning for automated stopping criterion in zero-shot learning:*  More importantly, a naive application of the partitioning scheme in [Yaman et al, 2020] to the zero-shot setting fails due to overfitting, since there is no stopping criterion [5]. Thus, an automated stopping criterion for zero-shot learning with broad applicability is critically needed. Our proposed framework is well-aligned with the use of a validation set in the conventional database-trained setting, where the bias-variance trade-off is used to motivate early stopping as regularization, and the stopping criterion is identified using a separate validation set.  Using the same bias-variance trade-off motivation, having a validation set can help us devise a stopping criterion, but this has not been feasible in other zero-shot learning approaches due to the partitioning framework they have used. Therefore, unlike existing self-supervised models in the zero-shot setting, our proposed ZS-SSL partitions measurements into three sets, where two sets are used for self-supervision and the last set is used for self-validation, where the validation set serves as a proxy for generalization error that balances the bias-variance trade-off with the proper stopping criterion. Hence, ZS-SSL tackles the challenge of the lack of validation set in zero-shot learning for MRI reconstruction, which may inspire further investigations in zero-shot learning applications in other domains.
> >
> >
> >
> > We kindly note that while it may seem straightforward in hindsight, we think that our partitioning framework for joint self-supervision and self-validation is a major innovation. In a way, this is similar to how holdout masking (or the partition into two sets) seems natural for self-supervision now, but it was entirely novel when first proposed.
> >
> >
> > Finally, we hope that we clarified these very thoughtful comments. We would be glad to discuss further.
> >
> > References:
> >
> > [Yaman et al, 2020] DOI: 10.1002/mrm.28378.
> >
> > [1] DOI: 10.1016/j.neuroimage.2015.10.019.
> > [2] DOI:10.1038/s41467-021-25431-8.
> > [3] DOI: 10.1002/jmri.25695.
> > [4] DOI: 10.1109/TMI.2021.3075856.
> > [5] DOI: 10.1109/EMBC44109.2020.9176241.

---

> > > ### Comment · Reviewer_pt6r · 2021-11-20
> > > **Follow-up on Figure 2**
> > >
> > > **Figure 2**
> > >
> > > Part1. Thanks for investigating K=100 as well. It occurs to the reviewer that K=100 with early stopping should give a significantly better results. How come the authors observed a similar performance for K=10,25,50,100 and decided to go with K=10 throughout the experiments? The reviewer notes this could be due to the failure of the metric to capture the perceived quality, but any other explanations by the authors would also be appreciated.
> > >
> > > Part2. The reviewer still finds the right-most plot misleading since it suggests that no benefit comes from ZS-SSL-TL (as the validation error is almost monotonically increasing).
> > >
> > > **Thanks for all the other clarifications.**

---

> > > > ### Author Response · Authors · 2021-11-22
> > > > **Further clarifications on Figure 2**
> > > >
> > > > #### **pt6r.C4:  Figure 2 - Part 1**
> > > >
> > > > We agree that K=100 converges/stops in fewer epochs, and that the metrics may not always reflect the true reconstruction quality in MRI. For the latter point, this is indeed why we provided visual experiments throughout the study, including the selection of K (Figure 9). We would like to note that for each K value the reconstruction is performed at the converged epoch selected by the stopping criterion. We observed that all the models evaluated at their converged epoch for K=10, 25, 50 and 100 have similar metrics and provide similar artifact-free reconstruction quality (Figure 9). Note even though K=100 stops at an earlier epoch, its per epoch cost is 10 times as much as K=10, and overall computational complexity remains similar for K=10, 25, 50 and 100.   Thus, we chose K= 10 for the remainder of the experiments since we wanted a fixed K, while acknowledging in the text that other K=25, 50 and 100 values can also be used (Sec. 4.4).
> > > >
> > > > #### **pt6r.C4:  Figure 2 - Part 2**
> > > > We thank the reviewer for further questions on this point. We would like to kindly note that if we zoom into the validation loss of ZS-SSL-TL closely, we see that the validation decreases until the breakpoint, for the first ~30 epochs, and then increases. We would like to note that we have 4 plots in the Figure due to space limitations, and while zooming in further to the validation curve would be desirable, this would cause inconsistencies in the presentation.
> > > >
> > > >
> > > > Finally, we appreciate the reviewer's continued effort in improving our work. Please let us know if we can clarify further.

---

> > ### Comment · Reviewer_pt6r · 2021-11-20
> > **Follow-up on the computational comparison and ZS-SSL-TL**
> >
> > The reviewer appreciates the comprehensive and useful response of the authors and all the changes made in a limited time.
> >
> > **Computational comparison**
> >
> > The reviewer finds Table 2 in the appendix of great value. However, is it possible that there is a mistake in computing the runtimes? Or maybe a lot of computational resources have been used (e.g., more than one GPU)? For example, typically, DIP-based methods require on the order of 10,000 gradient iterations to converge for a given fastMRI slice. This would amount to a runtime on the order of 1 hour using a single GPU.
> > Moreover, even traditional methods such as sparsity- or TV-based methods require on the order of 10-100 seconds to reconstruct a slice. Thus, one would expect more computational cost for CG-SENSE as well.
> >
> > **Table 2 (which is now Table 3)**
> >
> > Thanks for the clarification. There is a caveat which the authors can check easily. If ZS-SSL-TL takes way longer to converge for ``trained on brains & tested on knees,`` then it might be the case that at some point of test-time training, the network becomes a random prior and that's why further training on knees improves the reconstruction accuracy significantly and matches that of ``trained one knees & tested on knees.`` The authors can observe this as follows. Suppose ZS-SSL-TL is being run on the network pre-trained on brains. Then, if up to some epoch, the reconstruction error goes up and then suddenly starts to go down again, then the network has forgotten the learned prior of brains and is learning a new prior for knees.
> >
> > The reviewer remarks that if this happens, it is fine by nature because it improves generalization anyway, but then this implies that ZS-SSL-TL is not involved here, and all the improvements are merely achieved by ZS-SSL itself.
> >
> > **ZS-SSL has no prior domain information**
> >
> > The explanation given by the authors makes sense, though taking hyper-parameters into account would add more value. Of course, the reviewer understands that it's not feasible in the limited time left.

---

> > > ### Author Response · Authors · 2021-11-22
> > > **Further clarifications on computational comparison and ZS-SSL-TL**
> > >
> > > We thank the reviewer again for their continued efforts and interest in improving our work. We are also glad to hear that the reviewers found our response comprehensive.
> > >
> > > #### **pt6r.C1:  The computational comparisons**
> > >
> > > We thank the reviewer for the comment. We agree with the reviewer the original DIP method for image restoration tasks require thousands of iterations to converge [1]. However, these DIP methods are based on generative networks, starting from a random noise input and generating the image of interest. In contrast, in the physics-guided MRI reconstruction setting, DIP-recon starts directly from the acquired measurements $y_\Omega$ (Sec. 4.3). Hence, its convergence is achieved in fewer iterations as shown in Figure 8. We also checked the reconstructions after thousands of iterations. However, these suffer from major noise amplification. As discussed in our previous response, DIP-type MRI reconstructions use all acquired measurement indices $\Omega$ for both training and defining the loss, and since $\Omega$ is used in enforcing data consistency in the network, using $\Omega$ also for defining the loss essentially amounts to calculating the loss over residual artifacts \& noise. Hence, DIP-Recon leads to noise amplification and artifacts in the reconstructions. Therefore, the empirically selected 50 epochs leads to faster reconstructions, but with artifacts. We also note that our machines use 4 V100 GPUs, which is now included in the caption of Table 2.
> > >
> > > Regarding the traditional methods such as sparsity or TV-based methods, we would like to kindly note that these are now available in clinical MRI systems with minimal latency. The 10-100 seconds per slice quoted here may refer to earlier implementations of compressed sensing in research settings. Similarly, CG-SENSE computational time is even lower. This is not surprising since the physics-guided DL reconstruction uses ~100 CG iterations over its DC units and still has a computational time in the order of milliseconds for database-trained methods.
> > >
> > > #### **pt6r.C2:  Table 3**
> > > We appreciate the reviewers’ insightful comments on this point. Upon the reviewer suggestions, we investigated the reconstruction error of ZS-SSL-TL for trained on knees & tested on knees, as well as trained on brains & tested on knees, by computing both the PSNR and SSIM across epochs for the two cases. As the reviewer suggested, if the convergence were to a random prior first, then the reconstruction error is supposed to go up before suddenly dropping. In other words, one would expect PSNR and SSIM to decrease up to some epoch. However, this was not the case observed in the experiments. Rather, the experiments align with the expectation that PSNR \& SSIM gradually increase (i.e. reconstruction error gradually goes down) until the stopping condition and then starts to gradually decrease (reconstruction error goes up) afterwards.
> > >
> > >
> > > These observations align with the fact that we are initializing our model with a set of parameters optimized for MRI data. In terms of the priors that the reviewer alludes to, we do not expect the knee prior and brain prior to be very distinct though there are some differences. Thus, when we do subject-specific training, it is easier to move from the brain prior to the subject-specific knee prior than to move from the brain prior to a random prior first and then to the subject-specific knee prior. Furthermore, as we discussed in our first response, the in-domain pretrained model/prior enables faster convergence compared to the out-of-domain one, with slight fluctuations over the final metrics since both end up with good subject-specific models/priors.
> > >
> > >
> > > The observation about learned models/priors for knee and brain being not very different is similar to the usage of sparsity models in conventional MRI techniques. For example, wavelets can be used for both knee and brain MRI as a sparsifying transform, but respective thresholds may change in a data-dependent manner. In the zero-shot deep learning case, the same pretrained model can also be used as a starting point for both knee and brain MRI, while fine-tuning changes the parameters based on the data.
> > >
> > >
> > > #### **pt6r.C3:  ZS-SSL has no prior domain information**
> > >
> > > We agree with the reviewer taking different hyper-parameter into account may enable further improvements in the reconstruction quality. In replying to this comment, we also realized that we had not included this in the first revision. Upon the reviewer’s suggestion, we have now highlighted this case in the Conclusions section and uploaded the modified PDF.
> > >
> > > Thank you again for these insightful comments. Please let us know if we can clarify further.
> > >
> > > References:
> > > [1] DOI: 10.1109/CVPR.2018.00984.

---

### Author Response · Authors · 2021-11-17
**General Response to All Reviewers**

We would like to thank all the reviewers for insightful and constructive comments. We are delighted to see all reviewers found our paper well-written with clear explanation and extensive experiments. We are also happy to see the reviewers have generally well-received the strengths of our paper in tackling challenges associated with MRI, and its importance in integrating deep learning reconstruction into clinical practice.

We appreciate the reviewers' feedback and have addressed their comments in our response, and incorporated relevant changes to the text and uploaded the modified PDF. For their convenience, we have marked the changes in blue for easier tracking.


Following our response and the text changes, we hope that the reviewers readily find our paper as a valuable contribution to the ICLR subject area on applications, specifically in biomedical imaging and health care. We would be happy to engage in the further discussion if the reviewers have any additional questions.

Thank you,

the authors

---

### Decision · Program_Chairs · 2022-01-20

**Decision:**

Accept (Poster)

**Comment:**

The paper considers the problem of accelerated magnetic resonance imaging where the goal is to reconstruct an image from undersampled measurements. The paper proposes a zero-shot self-supervised learning approach for accelerated deep learning based magnetic resonance imaging. The approach partitions the measurements from a single scan into two disjoint sets, one set is used for self-supervised learning, and one set is used to perform validation, specifically to select a regularization parameter. The set that is used for self-supervised learning is then again split into two different sets, and a network is trained to predict the frequencies from one set based on the frequencies in the other set. This enables accelerated MRI without any training data.
The paper evaluates on the FastMRI dataset, a standard dataset for deep learning based MRI research, and the paper compares to a trained baseline and an un-trained baseline (DIP). The paper finds their self-supervised method to perform very well compared to both and shows images that indicate excellent performance. It would have been even better to compare the method on the test set of the FastMRI competition to have a proper benchmark comparison.

Here is how the discussion went:
- Reviewer pt6r is supportive of acceptance, but notes a few potential irregularities, such as the method pre-trained on brain and tested on knees performing better than the method pre-trained on knees and tested on knees, and not providing a comparison of the computational cost. The authors added a table to the appendix revealing that the computational costs are very high, much higher than for DIP even. The reviewer was content with the response and raised the score.

- Reviewer mBMk argues that the contribution is too incremental compared to prior work, in particular relative to the results of [Yaman et al., 2020], and also argues that the idea of partitioning the measurements is not new. The authors argue in response that their approach of partitioning the measurements is new, and the reviewer was inclined to raise the score slightly, but still thinks that the novelty on the technical ML side remains limited, and doesn't want to back the submission too much, and did not raise the score at the end in the system.

- Reviewer 19v3 has the concern that the all elements used (transfer learning, plug-and-play, etc) are well known techniques and have been applied before to MRI, and therefore thinks that the paper does not clear the bar for acceptance. The paper points out that while those ideas might be applied for the first time to MRI, they have been used before in other image reconstruction problems, in particular denoising.

I've read the paper in detail too, and am somewhat on the fence: I think it's very valuable to see that a clever application of self-supervised learning works so well for MRI. I agree with the reviewers that the technical novelty is relatively small, but on the other hand this is the first time that I see self-supervised learning being applied that successfully to MRI. I don't share the concern about novelty --- yes, the paper's approach builds on prior work, but it's not clear from the literature how well such a well tuned self-supervised learning approach would work.
What I would have liked to see in addition to the experimental results is a proper evaluation on the FastMRI dataset: An advantage of the FastMRI dataset is that it provides a benchmark and if researchers evaluate on that benchmark (on the testset/validation set) we can compare different methods well. The paper under review doesn't do that, it only evaluates on 30 test slices, and thus it's hard to benchmark the method. Also, the paper would benefit from more ablation studies.

In conclusion, I would be happy to discuss this paper at the conference, and think that other researchers in the intersection of deep learning and inverse problems would be too, and therefore recommend acceptance.